# Characteristics associated with viral suppression among HIV-infected children aged 0–14 years in Mozambique, 2019

**Neusa Vanessa Fernandes Abdul Fataha**[1]\*, **Sandra Gaveta**[2], **Jahit Sacarlal**[3], **Erika Valeska Rossetto**[4], **Cynthia Semá Baltazar**[2], **Timothy Allen Kellogg**[5]

**1** Mozambique Field Epidemiology and Laboratory Training Program, Mozambique, **2** National Institute of Health, Maputo, Mozambique, **3** Faculty of Medicine, University of Eduardo Mondlane, Maputo, Mozambique, **4** MassGenics, Assigned to Centers for Disease Control and Prevention, Maputo, Mozambique, **5** Institute for Global Health Sciences, University of California San Francisco (UCSF), San Francisco, California, United States of America

\* neusa.fataha@gmail.com

**Data Availability Statement:** All data are in the manuscript and supporting information files.

**Funding:** This study was possible due the financial support for FETP coordinated by the National

## Abstract

The human immunodeficiency virus (HIV) is a global public health problem, disproportionally affecting sub-Saharan African countries including Mozambique. In 2019, of 150,000 estimated HIV-infected children in Mozambique, only 95,080 were on antiretroviral treatment and 73% virally suppressed. The objective of this study was to determine the characteristics associated with viral suppression in children. A cross-sectional study was carried out using records of viral load samples from children aged 0 to 14 years old who underwent viral load tests in 2019 in Mozambique. Secondary analyses were conducted on data obtained from Data Intensive Systems and Applications (DISA) of children enrolled in health facilities who had viral load tests registered. Viral suppression was defined as the presence of less than 1,000 copies/ml of blood. Multivariate logistic regression analysis was used to evaluate the characteristics associated with viral suppression. Of the 33,559 viral load sample records analyzed, 53% (17,794/33,559) were female. The average patient age was 8 (sd ± 4) years old. About 44% (14,888/33,559) of the children had a suppressed viral load, with 55% (8,258/14,888) being female and 16% (2,319/14,888) belonging to the 1–4 years old age group. Characteristics associated with viral suppression were the age groups of 5–9 years [AOR = 1.73; 95% CI 1.34–2.23; p<0.001] and 10–14 years old [AOR = 1.92; 95% CI 1.50–2.48; p<0.001] versus < 1 year. Other factors such as living in Maputo City [AOR = 1.61; 95% CI 1.26–2.05; p <0.001] versus Tete Province were also associated with viral suppression. Factors such as being male [AOR = 0.83; 95% CI 0.80–0.87; p <0.001)], living in the provinces of Niassa [AOR = 0.75; 95% CI 0.56–0.99; p <0.003], Cabo Delgado [AOR = 0.77; 95% CI 0.60–0.99; p <0.045] and Zambezia [AOR = 0.72 (95% CI: 0.56–0.92, p<0.008)] versus Tete Province, or being on ART for 2–5 years [AOR = 0.72 (95% CI: 0.61–0.85, p<0.001)] versus 11–14 years were associated with not being virally suppressed. More than half of children did not achieve viral suppression. The odds of viral suppression were highest among children aged 5–14 years and among children living in Maputo city.

Institute of Health of Mozambique through the PEPFAR Cooperative Agreement 5U2GGH000080. The funders had no role in study design, data collection and analysis, decision to publish, or preparation of the manuscript.

**Competing interests:** The authors declare that they have no conflicts of interests.

Further research is needed to better understand the challenges in achieving viral suppression in children.

## Background

Increased access to prevention and treatment services for Human Immunodeficiency Virus (HIV) have contributed to higher rates of viral suppression, reduced risk of HIV transmission and improved quality of life for people living with HIV (PLHIV) [1]. Estimates from 2019 indicate that only 95,080 out of 150,000 children living with HIV in Mozambique were on antiretroviral treatment (ART) [2]. There were 8,200 deaths due to Acquired Human Immunodeficiency Syndrome (AIDS) in children aged 0–14 years in the country in 2019 [2].

HIV attacks the human body's immune system, specifically CD4 cells, weakening immunity [1]. High levels of viremia can contribute to HIV transmission, illness, and ultimately death by destroying immune cells and speeding up the progression of HIV infection to AIDS [3,4]. It is important to monitor viral load levels in all patients, including those who are on ART [5]. According to the HIV viral load algorithm implementation guidelines of the Ministry of Health in Mozambique, a viral load test with a result below 1,000 copies/ml of blood is considered to have achieved viral suppression. [6]. Monitoring viral load in children on ART using a viral load test is essential for clinical decision making, identifying risk groups, and determining progress toward the goals of the UNAIDS 95-95-95 cascade for PLHIV [7].

This study, carried out in Mozambique, aimed to estimate the rate of viral suppression in children and identify the characteristics associated with viral suppression in children on ART.

## Materials and methods

### Study design and population

A cross-sectional study was conducted using secondary data from viral load samples records collected from children aged 0–14 years who underwent viral load laboratory testing. We included in the study viral load samples records from HIV-positive children living in all provinces of Mozambique who had viral load testing done during 2019 and who started ART at different times.

### Data collection procedures and statistical analysis

Secondary data from DISA (Data Intensive Systems and Applications), an electronic laboratory management and sample forwarding system that covers all of Mozambique's provinces was used. This system allows samples to be registered at health facilities which send the information via the internet to the processing laboratory and then receive the results in electronic format.

DISA data are derived from information included on viral load test request and referral forms filled out by health facilities across Mozambique. Viral load tests were requested by medical staff for reasons such as routine follow-up and to confirm suspected treatment failure.

The DISA system is test-based, with each record representing an individual viral load test. Therefore, individual patients who tested more than one time in the same year could be represented multiple times.

Secondary data extracted included socio-demographic and clinical information of children who had viral load records tests in health facilities with ART. Independent variables included

in the analysis were gender, age, province, duration of ART and reasons for testing. Duration of ART for each observation was estimated by the difference between the date of sample collection and date of ART initiation.

The variable "Viral suppression" was created from the viral load values and took the category "1 = Yes" in individuals with a viral load less than 1,000 copies/ml blood and "0 = No" in individuals with a viral load equal to or greater than 1,000 copies/ml blood.

All analyses were conducted using Stata 16.0 (Stata Corp Software 16.0). Chi-square tests were used to determine associations between the independent categorical variables and viral suppression.

Univariate and multivariable logistic regression analysis were used to estimate the adjusted odds ratios (aOR) of continuous independent variables and viral suppression. Variables with outcomes with p values < 0.05 in the univariate analyses were included in the multivariate logistic regression model. The significance level adopted in the chi-square test and the univariate and multivariable logistic regression analyses was 0.05. Box plots were constructed to compare the distribution of HIV viral load logarithm among children by gender and age-group. If viral load results were classified as "undetectable," then the lowest limit of detection value was used for box plot analyses, typically 10, 20, or 40 copies/ml, depending on the laboratory device used.

### Ethical considerations

This study was approved by the NATIONAL COMMITTEE ON BIOETHICS FOR HEALTH with the reference **158/CNBS/13.**

## Results

### Baseline characteristics of study participants

The demographic and clinical characteristics of the participants are illustrated in Table 1. The mean age of the 33,559 children included in the analysis was 8 years (SD ± 4), 20% (6,858/ 33,559) were in the age group of 1–4 years old and 53% (17,797/33,559) were female. About 49% (16,456/33,559) of the study participants were on ART for 0–2 years and 81% (27,038/ 33,559) were tested as routine follow-up.

### Characteristics of the study population and viral suppression

Of a total of 33,559 viral load samples records from children aged 0–14 years on ART who had a viral load test., overall, about 44% (14,888 / 33,559) were virally suppressed. Viral suppression was higher for female children at 46% (8258/17794) than male children at 42% (6630/15765), highest for 10–14 year-olds at 50% (6484/13006) and lowest for 1–4 year-olds at 34% (2319/ 6858). Children living in Maputo city had the highest viral suppression at 58% (2562/4449), with children living in Nampula province with the lowest at 33% (1259/3841). Children on ART the longest time (11–14 years) had the highest suppression at 57% (348/613) and children on ART the shortest of time (0–2 years) had the lowest viral suppression at 43% (7003/16456). Viral suppression was higher for children who tested because of routine monitoring at 49% (13218/27038) compared with children who were being evaluated for treatment failure (1465/ 6086; 24%) (Table 2).

### Predictors of viral suppression and distribution of viral load

The associations between viral suppression in children and various sociodemographic and health-related variables are illustrated in Table 3. The results of the multivariable logistic

**Table 1. Characteristics of study participants, Mozambique–January to December 2019.**

| Characteristics | | Frequency N = 33,559 | % |
|---|---|---|---|
| **Age (years)** | | | |
| | < 1 | 289 | 1% |
| | 1–4 | 6,858 | 20% |
| | 5–9 | 13,406 | 40% |
| | 10–14 | 13,006 | 39% |
| **Sex** | | | |
| | Female | 17,794 | 53% |
| | Male | 15,765 | 47% |
| **Province** | | | |
| | Niassa | 812 | 2% |
| | Cabo Delgado | 2,051 | 6% |
| | Nampula | 3,841 | 12% |
| | Zambézia | 5,154 | 15% |
| | Tete | 283 | 1% |
| | Manica | 1,020 | 3% |
| | Sofala | 1,699 | 5% |
| | Inhambane | 2,534 | 8% |
| | Gaza | 5,777 | 17% |
| | Maputo Province | 5,939 | 18% |
| | Maputo City | 4,449 | 13% |
| **ART time (years)** | | | |
| | 0–2 | 16,456 | 49% |
| | 3–5 | 9,978 | 30% |
| | 6–10 | 6,512 | 19% |
| | 11–14 | 613 | 2% |
| **Reasons for test** | | | |
| | Routine | 27,038 | 81% |
| | After breastfeeding | 26 | 0% |
| | Suspected of Treatment Failure | 6,086 | 18% |
| | Non specific | 293 | 1% |
| | Not reported | 116 | 0% |

regression analysis showed that the characteristics associated with viral suppression were the age groups of 5–9 years [AOR = 1.73; 95% CI 1.34–2.23; p <0,001] and 10–14 years [AOR = 1.92; 95% CI 1.50–2.48; p <0.001] versus < 1 year old. Other factors such as living in Maputo City [AOR = 1.61; 95% CI 1.26–2.05; p <0.001] versus Tete Province were also associated with viral suppression. Doing a viral load test for reasons of routine follow-up (versus reasons relating to testing after breastfeeding) was also associated with viral load suppression [AOR = 3.01; 95% CI 2.82–3.22; p <0.001].

Factors such as being male [AOR = 0.83; 95% CI 0.80–0.87; p <0.001)], living in the provinces of Niassa [AOR = 0.75; 95% CI 0.56–0.99; p <0.040], Cabo Delgado [AOR = 0.77; 95% CI 0.60–0.99; p <0.045], or Zambezia [AOR = 0.72 (95% CI: 0.56–0.92, p<0.008)] versus Tete Province, or being on ART for less than 2 years [AOR = 0.80; 95% CI 0.67–0.95; p <0.013] and for 2–5 years [AOR = 0.72 (95% CI: 0.61–0.85, p <0.001)] versus 11–14 years were associated with an unsuppressed viral load.

Fig 1 presents a box plot with HIV viral load values in children aged 0–14 years on a logarithmic scale of 10. The horizontal lines represent the median viral load values for each age

**Table 2. Study population characteristics and viral suppression, Mozambique– 2019.**

| Characteristics | Total | Viral suppression (<1000 copies/ml) | | Viral suppression (%) | p-value |
|---|---|---|---|---|---|
| | | Yes (%) | No (%) | | |
| **Overall** | 33,559 | 14,888 | 18,671 | 44% | – |
| **Gender** | | | | | <0.001 |
| Male | 15,765 | 6,630 (45%) | 9,135 (49%) | 42% | |
| Female | 17,794 | 8,258 (55%) | 9,536 (51%) | 46% | |
| **Age (years)** | | | | | <0.001 |
| <1 | 289 | 100 (1%) | 189 (1%) | 35% | |
| 1–4 | 6,858 | 2,319 (16%) | 4,539 (24%) | 34% | |
| 5–9 | 13,406 | 5,985 (40%) | 7,421 (40%) | 45% | |
| 10–14 | 13,006 | 6,484 (43%) | 6,522 (35%) | 50% | |
| **Province** | | | | | <0.001 |
| Niassa | 812 | 281 (2%) | 531 (3%) | 35% | |
| Cabo Delgado | 2,051 | 766 (5%) | 1,285 (7%) | 37% | |
| Nampula | 3,841 | 1,259 (9%) | 2,582 (14%) | 33% | |
| Zambézia | 5,154 | 1,846 (12%) | 3,308 (18%) | 36% | |
| Tete | 283 | 136 (1%) | 147 (0,8%) | 48% | |
| Manica | 1,020 | 450 (3%) | 570 (3%) | 44% | |
| Sofala | 1,699 | 758 (5%) | 941 (5%) | 45% | |
| Inhambane | 2,534 | 1,087 (7%) | 1,447 (8%) | 43% | |
| Gaza | 5,777 | 2,609 (18%) | 3,168 (17%) | 45% | |
| Maputo Province | 5,936 | 3,134 (21%) | 2,805 (15%) | 53% | |
| Maputo City | 4,449 | 2,562 (17%) | 1,887 (10%) | 58% | |
| **ART time (years)** | | | | | <0.001 |
| 0–2 | 16,456 | 7,003 (43%) | 9,453 (51%) | 43% | |
| 3–5 | 9,978 | 4,258 (29%) | 5,720 (31%) | 43% | |
| 6–10 | 6,512 | 3,279 (22%) | 3,233 (17%) | 50% | |
| 11–14 | 613 | 348 (2%) | 265 (1%) | 57% | |
| **Reasons for requesting a viral load test** | | | | | <0.001 |
| Routine | 27,038 | 13,218 (89%) | 13,820 (74%) | 49% | |
| After breastfeeding | 26 | 8 (0,05%) | 18 (<1%) | 31% | |
| Suspected of Treatment Failure | 6,086 | 1,465 (10%) | 4,621 (25%) | 24% | |
| Non specific | 293 | 153 (1%) | 140 (0,7%) | 52% | |
| Not reported | 116 | 44 (0,3%) | 72 (0,3%) | 38% | |

group. The variation in viral load was similar for males and females in all age groups. Children aged 6 to 14 years, for both sexes, had the lowest median viral load.

## Discussion

In this study of children 0–14 years on ART, the prevalence of viral suppression (<1000 copies/ml) was 44%, substantially lower that the national target of 90% suppression of children who have initiated ART.

This result was compared to a study in South Africa by Geoffrey et al. which reported a viral suppression prevalence of 66% among children and adolescents aged 0–16 years on ART [8]. We posit that this difference in viral suppression rates has to do with the fact that the South African study included children above 14 who may have been able to better adhere to their ART treatment regimen.

**Table 3. Predictors for viral suppression in children on ART (univariate and multivariate logistic regression), Mozambique– 2019.**

| Characteristics | n | Univariate | | Multivariable | |
|---|---|---|---|---|---|
| | | OR[a] (95% CI) | p-value | AOR[b] (95% CI) | p-value |
| **Gender** | | | | | |
| Male | 15,765 | 0.83 (0.80–0.87) | <0.001 | 0.83 (0.80–0.87) | <0.001 |
| Female | 17,794 | Ref | | Ref | |
| **Age (years)** | | | | | |
| <1 | 289 | Ref | | Ref | |
| 1–4 | 6,858 | 0.97 (0.75–1.23) | 0.782 | | |
| 5–9 | 13,406 | 1.52 (1.19–1.95) | <0.001 | 1.73 (1.34–2.23) | <0.001 |
| 10–14 | 13,006 | 1.87 (1.47–2.40) | <0.001 | 1.92 (1.50–2.48) | <0.001 |
| **Province** | | | | | |
| Niassa | 812 | 0.57 (0.43–0.75) | <0.001 | 0.75 (0.56–0.99) | 0.040 |
| Cabo Delgado | 2,051 | 0.64 (0.50–0.83) | <0.001 | 0.77 (0.60–0.99) | 0.045 |
| Nampula | 3,841 | 0.53 (0.41–0.67) | <0.001 | 0.81 (0.64–1.05) | 0.108 |
| Zambezia | 5,154 | 0.60 (0.47–0.77) | <0.001 | 0.72 (0.56–0.92) | 0.008 |
| Tete | | Ref | | Ref | |
| Manica | 1,020 | 0.85 (0.66–1.11) | 0.239 | | |
| Sofala | 1,699 | 0.87 (0.68–1.12) | 0.282 | | |
| Inhambane | 2,534 | 0.81 (0.63–1.03) | 0.097 | | |
| Gaza | 5,777 | 0.89 (0.70–1.13) | 0.345 | | |
| Maputo Province | 5,939 | 1.20 (0.95–1.53) | 0.121 | | |
| Maputo City | 4,449 | 1.47 (1.15–1.87) | 0.002 | 1.61 (1.26–2.05) | <0.001 |
| **ART time(years)** | | | | | |
| <2 | 16,456 | 0.57 (0.49–0.68) | <0.001 | 0.80 (0.67–0.95) | 0.013 |
| 3–5 | 9,978 | 0.56 (0.47–0.66) | <0.001 | 0.72 (0.61–0.85) | <0.001 |
| 6–10 | 6,512 | 0.77 (0.66–0.91) | | | |
| 11–14 | 613 | Ref | | Ref | |
| **Reasons for requesting a viral load test** | | | | | |
| After breastfeeding | | Ref | | Ref | |
| Routine | 27,038 | 2.15 (0.93–4.95) | 0.071 | 3.01 (2.82–3.22) | <0.001 |
| Suspected of Treatment Failure | 6,086 | 0.71 (0.30–1.64) | 0.428 | | |
| Non specific | 293 | 2.46 (1.03–5.83) | 0.041 | 3.41 (2.6–4.34) | <0.001 |

[a]OR- Odds ratio.

[b]AOR- Adjusted Odds ratio.

The results of our study showed that as age increases, the chance of achieving viral suppression also increases. This finding was similar to a study conducted in Ethiopia in 2016–2019 by Diress et al. involving 235 adults and children which showed an association between viral suppression and age and that the likelihood of viral suppression increased with the age [9].

Low viral suppression among younger children may be linked to lack of adherence, ART formulations, or caregiver-related factors [10]. Adequate adherence to ART contributes to lower viral loads and reduced viral resistance. Many studies have discussed ART formulations for children with HIV, including swallowing large tablets. Caregivers should follow recommendations such as opening capsules, combining medication with food and knowing which tablets are water-dispersible, which can make ART adherence caregiver-related [11].

Caregiver-related factors such as delayed HIV disclosure in children will hinder children's opportunity to understand and accept their illness, leaving the child confused about what they need to do as they grow up [12].

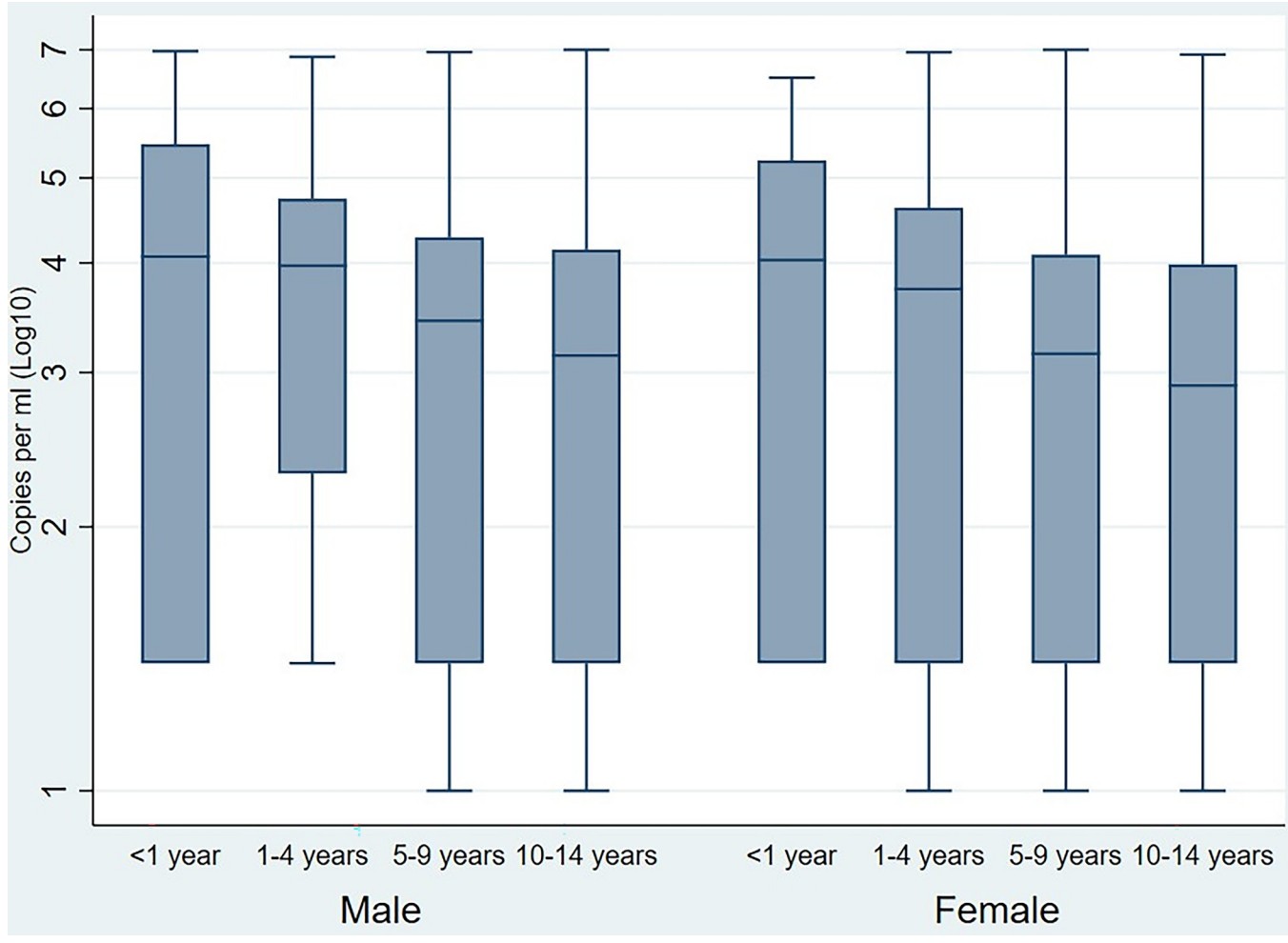

**Fig 1. Logarithm HIV viral load in children by sex and age, Mozambique– 2019.**

A study carried out in India showed that there are still challenges in disclosing HIV status in children and adolescents. The majority of caregivers wanted their children to know their HIV status when the child was at least 13 years old or older [13].

We also found that males had greater odds of not being virally suppressed than females. A study carried out in Kenya in 2014–2015 with children aged 0 to 15 years old also showed that being male children was associated with viral suppression [14]. A study involving adults, adolescents and children receiving ART in Cameroon showed that men achieved viral suppression of 80.9% and women of 75.9% respectively [15]. Although the study reported relatively greater viral suppression in men than women, data from many studies suggest that men are more likely to experience virologic failure than women. This can be explained by behavioral patterns related to risk, such as alcohol and drug use and lower acceptance of health services [16].

Our findings showed that children from Maputo City had higher chances of achieving viral suppression than children from other regions. This may be explained by the fact that the region allows for easier access to the health facilities.

The inability of many children to maintain viral suppression may be explained in part by the length of time that they are on ART [17]. In this study, it was found that longer duration of ART use was associated with non-viral suppression. Esther et al. found that among children

and adolescents receiving intensive ART adherence advice in Uganda, as ART duration increased, the proportion of children suppressed up to 14 years old also increased [18]. The Uganda study found the opposite of the trend identified in this study. Possible factors to achieve viral suppression may be linked to greater adh-erence to ART by older children. General adherence can also be influenced by the complexity of the treatment regimen, responsibility, knowledge about the disease and the importance of ART for viral load suppression, receipt of community-based adherence support [19].

## Limitations

This study was cross-sectional in design and did not follow children over time to measure lost-to-follow up, baseline clinical characteristics, or document transfers of children to other facilities. One weakness of this study was the use of secondary data which restricted data analysis and interpretation to only variables that were captured in DISA patient records. It is possible that some other variables that were not captured, such as income, distance to the health facilities, adherence to ART, and line of therapy, could also be important predictors for viral load suppression. In addition, the data in this study were based on viral load test results, so a child with more than one test per year may have been represented more than once.

## Conclusions

More than half of the children studied who were on ART did not achieve viral suppression. The odds of viral suppression increased with age and were higher for children living in Maputo City versus other Mozambican provinces. Further research is needed to better understand the challenges in achieving viral suppression in children.

## Supporting information

**S1 Checklist.**
(DOCX)

**S1 File.**
(DOCX)

## Acknowledgments

We gratefully acknowledge the support received by the HIV Observation Platform from the National Health Observatory for permission to use data. Our special thanks is also extended to Arianna Unger and Peter Young from Centers for Disease Control and Prevention Mozambique.

## Author Contributions

**Conceptualization:** Neusa Vanessa Fernandes Abdul Fataha.

**Data curation:** Neusa Vanessa Fernandes Abdul Fataha.

**Formal analysis:** Neusa Vanessa Fernandes Abdul Fataha, Timothy Allen Kellogg.

**Investigation:** Neusa Vanessa Fernandes Abdul Fataha.

**Methodology:** Neusa Vanessa Fernandes Abdul Fataha, Timothy Allen Kellogg.

**Project administration:** Neusa Vanessa Fernandes Abdul Fataha.

**Resources:** Neusa Vanessa Fernandes Abdul Fataha.

**Supervision:** Sandra Gaveta, Jahit Sacarlal, Erika Valeska Rossetto, Cynthia Semá Baltazar, Timothy Allen Kellogg.

**Validation:** Jahit Sacarlal.

**Writing – original draft:** Neusa Vanessa Fernandes Abdul Fataha.

**Writing – review & editing:** Neusa Vanessa Fernandes Abdul Fataha.

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
