## [Decision Letter · Decision Letter 0]

13 Feb 2024

PONE-D-23-33823Characteristics associated with viral suppression among HIV-infected children aged 0-14 years in Mozambique, 2019PLOS ONE

Dear Dr. Fataha,

Thank you for submitting your manuscript to PLOS ONE. After careful consideration, we feel that it has merit but does not fully meet PLOS ONE’s publication criteria as it currently stands. Therefore, we invite you to submit a revised version of the manuscript that addresses the points raised during the review process.

We look forward to receiving your revised manuscript.

Kind regards,

Orvalho Augusto, MD, MPH

Academic Editor

PLOS ONE

 [This study was possible due the financial support for FETP coordinated by the National Institute of Health of Mozambique through the PEPFAR Cooperative Agreement 5U2GGH000080.].  

5. Please provide a complete Data Availability Statement in the submission form, ensuring you include all necessary access information or a reason for why you are unable to make your data freely accessible. If your research concerns only data provided within your submission, please write "All data are in the manuscript and/or supporting information files" as your Data Availability Statement.

6. Please remove your duplicate figure in your file inventory (Supporting Information).

Additional Editor Comments:

This is an important report on viral suppression among children and adolescents using a large and nationally representative dataset of viral load testing in Mozambique during 2019. The data reported here is informative and can help shape better policies for HIV control. Few issues:

1. Revise the aims statement. To me, there are 2 aims here: first is the reporting of the level of viral suppression. The second is finding the factors. This second aim is quite limited, given that you have a narrowed set of variables. Therefore, not including the first aim removes the relevance of this report.

2. How is DISA related to the entire information system in Mozambique? Are all samples collected for viral load in these age groups in Mozambique end up in DISA? Please provide more description on this.

3. About the time on ART variable. Children with less than 6 months of ART should not be mixed with other children. So, on every table, including the ART time variable, please subdivide the 0-2 years into 0-5 months, 6 to 11 months, and 12 to 23 months at least.

4. In tables 1 to 3, what is the difference between "non-specific" and "not reported categories" in the reasons for requesting a viral load test.

5. In the results section:

- Please describe the odds ratios. It is not enough to say something is associated. We need the magnitude and direction of the association.

- As a sensitivity analysis, another analysis like in table 2 should be added to the supplements excluding children with less than 6 months of ART.

- The current analysis in table 2 is problematic. First, why was Tete province chosen as a reference? A justification is needed. Second, why did some province dummy codes disappear on the adjusted analysis?

- Please change from "multivariate" to "multivariable."

6. About the log10 boxplots:

- What was done for undetectable viral loads?

- What was done if a viral load came with a zero value?

Reviewers' comments:

Reviewer's Responses to Questions

**Comments to the Author**

1. Is the manuscript technically sound, and do the data support the conclusions?

Reviewer #1: Yes

2. Has the statistical analysis been performed appropriately and rigorously? 

Reviewer #1: Yes

3. Have the authors made all data underlying the findings in their manuscript fully available?

Reviewer #1: Yes

4. Is the manuscript presented in an intelligible fashion and written in standard English?

Reviewer #1: Yes

5. Review Comments to the Author

Reviewer #1: Congratulations to the authors for an informative study. I have the following comments:

1) In the abstract result section: ''...or being on ART for 2-5 years [AOR= 0.72 (95% CI: 0.61-0.85, p<0.001)] versus 11-14 years were associated with not being virally suppressed''  does the years refer to age groups of the children, or the number of years on ART? The children included in the study are aged 0 to 14 years.

2) Methods section under study population: Please correct this sentence ''Therefore, individual patients who tested more than one time in the same year 'were be' represented multiple times.''

3)Could you please clarify under data collection procedures, where the secondary clinical data was extracted from - whether it is from the VL request form or from the HIV program database?

4) Could you provide the IRB approval reference number in the ethics consideration section?

5) The discussion section could be improved - i think there could be more discussion around the lack of child friendly formulations for children, challenges of status disclosure in children and adolescents, lack of child and adolescent friendly corners or support groups...and how all this could affect the viral suppression.

6. PLOS authors have the option to publish the peer review history of their article (what does this mean?). If published, this will include your full peer review and any attached files.

Reviewer #1: No

---

## [Author Response · Author response to Decision Letter 0]

7 May 2024

Additional Editor Comments:

1. Objectives - The objectives have been corrected; the first objective has been added as suggested by the reviewer. 

2. Descriptions of DISA - More descriptions of DISA have been provided. 

3. Variable duration of ART - It will not be possible to assess this variable using the suggested category because the variable duration of ART is presented in years in the database and not in months. 

4. Difference between "non-specific" and "not reported" categories in the reasons for requesting the viral load test meaning: 

Non-specific is a category used in the laboratory request when specimens submitted to laboratory for processing.

Not reported in this analysis mean that the reason for the viral load test was not completed or left blank in the lab request. Therefore, we interpret this as to mean the reason was “unknown”.

5. Results: 

- The magnitude and direction of the association are described in the discussion section.

- Variable duration of ART - as mentioned, it will not be possible to assess this variable using the suggested category because the variable duration of ART is presented in years in the database and not in months. 

- Tete province was chosen as a reference because it had the lowest number of children in the population and in table 2 the result of whether or not children were suppressed was close. 

- The text and tables have been updated from "multivariate" to "multivariable". 

6. About the log10 box plots: 

- For the box plots, if the patient was undetectable the lowest limit was used for VL box plot. There was no zero values in the data.

1. Reviewer no. 1:

1) The years refer to the number of years on ART 

2) The sentence ''Therefore, individual patients who tested more than once in the same year 'were' represented multiple times.'' has been corrected

3) It has been clarified in the data collection procedures from where the secondary clinical data was extracted. 

4) The IRB approval reference number was provided in the ethical considerations section. 

5) The discussion section was improved.

---

## [Decision Letter · Decision Letter 1]

30 May 2024

Characteristics associated with viral suppression among HIV-infected children aged 0-14 years in Mozambique, 2019

PONE-D-23-33823R1

Dear Dr. Abdul Fataha,

We’re pleased to inform you that your manuscript has been judged scientifically suitable for publication and will be formally accepted for publication once it meets all outstanding technical requirements.

Kind regards,

Jason T. Blackard, PhD

Academic Editor

PLOS ONE

Additional Editor Comments (optional):

None

Reviewers' comments:

Reviewer's Responses to Questions

**Comments to the Author**

1. If the authors have adequately addressed your comments raised in a previous round of review and you feel that this manuscript is now acceptable for publication, you may indicate that here to bypass the “Comments to the Author” section, enter your conflict of interest statement in the “Confidential to Editor” section, and submit your "Accept" recommendation.

Reviewer #1: All comments have been addressed

Reviewer #2: All comments have been addressed

2. Is the manuscript technically sound, and do the data support the conclusions?

Reviewer #1: Yes

Reviewer #2: (No Response)

3. Has the statistical analysis been performed appropriately and rigorously? 

Reviewer #1: Yes

Reviewer #2: Yes

4. Have the authors made all data underlying the findings in their manuscript fully available?

Reviewer #1: Yes

Reviewer #2: Yes

5. Is the manuscript presented in an intelligible fashion and written in standard English?

Reviewer #1: Yes

Reviewer #2: Yes

6. Review Comments to the Author

Reviewer #1: I have no further comments. All comments have been addressed fully and the manuscript has been improved.

Reviewer #2: The authors have addressed all the reviewer comments and no further comments from me. I therefore recommend that the manuscript be accepted for publication

7. PLOS authors have the option to publish the peer review history of their article (what does this mean?). If published, this will include your full peer review and any attached files.

Reviewer #1: No

Reviewer #2: No

---

## [Editor Report · Acceptance letter]

14 Jun 2024

PONE-D-23-33823R1 

PLOS ONE

Dear Dr. Fataha, 

I'm pleased to inform you that your manuscript has been deemed suitable for publication in PLOS ONE. Congratulations! Your manuscript is now being handed over to our production team.

Kind regards, 

on behalf of

Dr. Jason T. Blackard 

Academic Editor

PLOS ONE